# Inhibition of Kv2.1 Potassium Channels by MiDCA1, A Pre-Synaptically Active PLA_2_-Type Toxin from *Micrurus dumerilii carinicauda* Coral Snake Venom

**DOI:** 10.3390/toxins11060335

**Published:** 2019-06-12

**Authors:** Niklas Schütter, Yuri Correia Barreto, Vitya Vardanyan, Sönke Hornig, Stephen Hyslop, Sérgio Marangoni, Léa Rodrigues-Simioni, Olaf Pongs, Cháriston André Dal Belo

**Affiliations:** 1Institute for Cellular Neurophysiology, Center for Integrative Physiology and Molecular Medicine (CIPMM), University of the Saarland, D-66421 Hamburg, Germany; niklas.schuetter@googlemail.com (N.S.); oupon@t-online.de (O.P.); 2Interdisciplinary Centre for Research in Biotechnology (CIPBiotec), Federal University of Pampa (UNIPAMPA), Campus São Gabriel, São Gabriel 97300-000, RS, Brazil; barreto78@outlook.com; 3Molecular Neuroscience Group, Institute of Molecular Biology NAS RA, Hastratyan 7, Yerevan 0014, Armenia; vvardanyan@yahoo.com; 4Center for Molecular Neurobiology Hamburg, Experimental Neuropediatrics, UKE Hamburg, 20251 Hamburg, Germany; soenke.hornig@zmnh.uni-hamburg.de; 5Department of Pharmacology, Faculty of Medical Sciences, State University of Campinas (UNICAMP), Rua Tessália Vieira de Camargo, 126, Cidade Universitária Zeferino Vaz, Campinas 13083-970, SP, Brazil; hyslop@fcm.unicamp.br (S.H.); simioni@unicamp.br (L.R.-S.); 6Department of Biochemistry, Institute of Biology, State University of Campinas (UNICAMP), Rua Monteiro Lobato, 255, Cidade Universitária Zeferino Vaz, Campinas 13083-862, SP, Brazil; marango@unicamp.br

**Keywords:** *Micrurus dumerilii carinicauda* venom, phospholipase A_2_ neurotoxin, mouse dorsal root ganglion neurons, Kv2 selective inhibition

## Abstract

MiDCA1, a phospholipase A_2_ (PLA_2_) neurotoxin isolated from *Micrurus dumerilii carinicauda* coral snake venom, inhibited a major component of voltage-activated potassium (Kv) currents (41 ± 3% inhibition with 1 μM toxin) in mouse cultured dorsal root ganglion (DRG) neurons. In addition, the selective Kv2.1 channel blocker guangxitoxin (GxTx-1E) and MiDCA1 competitively inhibited the outward potassium current in DRG neurons. MiDCA1 (1 µM) reversibly inhibited the Kv2.1 current by 55 ± 8.9% in a *Xenopus* oocyte heterologous system. The toxin showed selectivity for Kv2.1 channels over all the other Kv channels tested in this study. We propose that Kv2.1 channel blockade by MiDCA1 underlies the toxin’s action on acetylcholine release at mammalian neuromuscular junctions.

## 1. Introduction

Voltage-gated potassium (Kv) channels play an important role in controlling neuronal excitability [1,2]. Inhibition of these channels leads to membrane depolarization and may result in increased neurotransmitter release at nerve terminals [3,4,5]. In the peripheral nervous system, Kv channel inhibition is associated with hyperexcitability, tissue paralysis, and potential neuronal cell death. Many venomous animals, such as marine cone snails, spiders, scorpions, sea anemones and snakes produce toxins that act on Kv channels to facilitate the capture of prey [6,7,8]. Snake venom phospholipases A_2_ (sPLA_2_) constitute a toxin subfamily with dual activity. On the one hand, sPLA_2_-type toxins exhibit hydrolytic activity towards lipids, while on the other hand some of these toxins block neuronal Kv channels in a tissue-specific manner [9,10,11,12]. MiDCA1 is an Asp49 PLA_2_ β-neurotoxin present in the venom of the coral snake *Micrurus dumerilii carinicauda* [13,14]. Experiments with mouse hemidiaphragm preparations have shown that, like other neurotoxic PLA_2_s [15,16], MiDCA1 affects neurotransmitter release and the corresponding muscle twitch tension [13].

The precise mechanism of action of MiDCA1 is still unknown, although MiCDA1-induced hydrolysis of phosphatidylcholine to lysophosphatidylcholine and fatty acid could potentially underlie the toxin´s effect on neurotransmitter release [16,17]. MiDCA1 may also affect tissue excitability through direct interaction with Kv channels. In this study, we examined this hypothesis by investigating the effect of MiDCA1 on Kv channels in primary cultures of mouse dorsal root ganglion (DRG) neurons and in a *Xenopus* oocyte heterologous expression system. Our results show that MiDCA1 inhibits Kv2.1 potassium channels. 

## 2. Results

In agreement with a previous report [18], the stepwise depolarization of cultured mouse DRG neurons evoked a series of outward currents that activated slowly, rapidly reached a plateau phase, and then inactivated (Figure 1A). A plot of the normalized peak outward current (I_norm_) against test voltages showed a nonlinear current increase between –30 mV and 0 mV followed by an almost linear current increase at test voltages ≥ 0 mV (Figure 1D).

To examine the effect of MiDCA1 on outward current at +60 mV (Figure 1), the preparations were perfused with 1 µM MiDCA1 prior to applying a depolarization pulse. At this concentration, MiDCA1 inhibited outward current (Figure 1B). Since no further inhibition was observed with the application of a higher concentration (2.4 µM) of MiDCA1 in DRG neurons (data not shown), we chose to use a concentration of 1 μM to economize on the limited amount of toxin available. Of the 16 DRG neurons from four preparations, the outward currents from nine neurons showed marked inhibition (41 ± 3%, *p* < 0.005) upon MiDCA1 application. These DRG neurons were arbitrarily classified as MiDCA1-sensitive since, in the remaining seven DRG neurons, the initial current decrease after MiDCA1 application was marginal (10 ± 3% inhibition; *p* > 0.05); the latter neurons were classified as MiDCA1-insensitive.

To obtain the MiDCA1-sensitive current, we subtracted the current recorded three min after the application of MiDCA1 from the control current (Figure 1C) in MiDCA1-sensitive neurons. The MiDCA1-sensitive current corresponded to a slowly inactivating outward current. Fitting a single exponential to the activation time course yielded a time constant of τ_act_ = 2.3 ± 0.2 ms (*n* = 5) at +60 mV. The current-voltage (I-V) relation for MiDCA1-sensitive and -insensitive currents was similar (Figure 1D). The I-V relation and kinetics of MiDCA1-sensitive current were reminiscent of those described for the K^+^ current mediated by Kv2 channels in DRG [18,19], suggesting that MiDCA1 targets Kv2 channels. To examine this possibility, we compared the MiDCA1-sensitive current with guangxitoxin (GxTx)-sensitive currents since GxTx is a well-established Kv2-specific channel gating modifier from the venom of the tarantula spider *Plesiophrictus guangxiensis* [7,20] (Figure 2). MiDCA1-sensitive DRG neurons (9 out of 16) responded to subsequent GxTx application with a small additional decrease in plateau current amplitude that was not significant (41 ± 3% vs. 51 ± 3%, *p* > 0.05) (Figure 2A,C,D). MiDCA1-insensitive DRG neurons also responded to subsequent GxTx applications with a small decrease in current (7 ± 8%; *p* > 0.5) (Figure 2C,D). We subsequently reversed the order of toxin application by applying GxTx first and then MiDCA1 (Figure 2B). GxTx application blocked DRG Kv currents to a similar degree (36 ± 4% inhibition; *n* = 5; *p* < 0.01) as MiCDA1 (Figure 2C). Subsequent application of MiDCA1 evoked a non-significant increase in current inhibition (17 ± 9%; *p* > 0.05) (Figure 2B–D). The extent of Kv2 blockade by GxTx was not altered by a four-fold increase in the concentration of this toxin (Figure 2E), suggesting that the residual current was probably mediated by other Kv subtypes not blocked by GxTx. In summary, the MiDCA1-sensitive subpopulation of DRG neurons was also sensitive to GxTx and vice versa. Based on these findings, we hypothesized that MiDCA1, like GxTx, targets Kv2 channels in DRG neurons.

We next expressed Kv2.1 channels in a *Xenopus* oocyte heterologous system to examine the inhibition by MiDCA1. The application of 1 μM MiDCA1 reversibly inhibited Kv2.1-mediated current by 55 ± 8.9% (Figure 3A,B). To check the selectivity of inhibition, we also applied the same concentration of toxin to Kv1.1, Kv1.2, Kv1.3, Kv1.4, Kv1.6, and KCNQ2/KCNQ3 voltage-dependent channels expressed in oocytes. None of these channel subtypes was inhibited by MiDCA1 (Figure 3C). Other channels such as Kv 3.1, Kv 4.2, M-channel, HERG, KCNQ1, BKα, and BKβ were also tested, but none of them was inhibited by 1 μM MiDCA1 (data not shown). These findings indicate that MiDCA1 showed selectivity for Kv2.1 over all the other Kv channels tested in this study. The limited amount of toxin available precluded similar experiments with other Kv channel subtypes, Kv2.2 channels, and heteromeric channels composed of Kv2 and silencing KvS subunits.

## 3. Discussion

Members of the Kv2 subfamily (Kv2.1, Kv2.2) are expressed in nerve tissues and have various splice variants. These proteins form heteromultimers with the so-called γ- or silent subunits of the Kv5, Kv6, Kv8, and Kv9 subfamilies [21]. DRG neurons express Kv2 α-subunits as well as several of Kv2 homotetramers and silencing Kv subunits (KvS) that heterotetramerize with Kv2 subunits [18], with the latter affecting Kv2 channel properties and pharmacology [22,23,24,25,26,27,28]. Since it is currently unknown which subunit combination of Kv2 delayed-rectifier channels is expressed in which DRG neuron type, it also remains to be determined which heteromultimeric Kv2 channel types represent the MiDCA1 target.

The secretogogue activity of snake venom PLA_2_ neurotoxins that disrupts synaptic function includes the production of docosahexaenoic acid (DHA) [29]. DHA acts at an extracellular site on neurons to produce a voltage- and time-dependent block of the delayed rectifier current (IK) [30]. Interestingly, the GxTx blockade of Kv2 channels is also voltage-dependent [7,20]. GxTx acts as a gating modifier that apparently inhibits Kv2 channels by shifting the channel activation to very positive potentials. In contrast, many toxins, e.g., scorpion, bee, and sea anemone toxins [7,19,28,31,32,33], inhibit Kv channels by occluding the extracellular pore entrance. Whether MiDCA1 acts as a Kv2-channel gating modifier in a manner similar to GxTx will be an interesting subject for future studies.

## 4. Materials and Methods 

MiDCA1 was purified from *M. d. carinicauda* venom (obtained from Sigma Chemical Co., St. Louis, MO, USA) using reverse-phase high performance liquid chromatography (RP-HPLC), as previously described [13]. The purified toxin was quantified by the dye-binding assay of Bradford [34], and the concentration used in the experiments was calculated based on the molecular mass of the purified toxin. Guangxitoxin-1E was purchased from Alomone Laboratories (Jerusalem, Israel). Toxin solutions were prepared daily by dilution in saline immediately before use. Small to medium-sized neurons were isolated from the DRG of nine-week-old male C57Bl/6j mice with enzymatic and mechanical methods, as described previously [4,18]. DRG neurons were plated on laminin/poly-L-lysine/borate-coated cover-slips at a low density (2000–3000 per dish) and were maintained 1–3 days at 37 °C for recording K^+^ outward currents in physiological saline solution in the whole-cell patch-clamp configuration at 37 °C. Contamination of the recorded current by sodium and calcium currents was suppressed by adding 30 nM tetrodotoxin (TTX) and 1 µM nifedipine, respectively. Currents were evoked from a holding potential of −80 mV. We included a 2 s pre-pulse of −30 mV to inactivate rapidly inactivating Kv currents, e.g., those expressed by members of the Kv3 and Kv4 family. Thereafter, we applied 1 s voltage steps to test potentials between −30 mV and +60 mV in 15 mV increments.

The expression of various Kv channels in *Xenopus* oocytes and their recordings were done as previously described [35]. Briefly, oocytes were surgically removed and defolliculated with collagenase (3 μg/μL) in OR2 solution. After a 24 h incubation in OR2 solution supplemented with 1.8 mM Ca^2+^ and gentamicin (50 μg/mL), the oocytes were injected (50 nL/oocyte) with mRNA encoding the desired potassium channels using a Nanoject2000 microinjector (Drumont Scientific Co., Broomall, PA, USA). mRNA was synthesized using a T7 mMessage mMachine kit (Ambion, Austin, TX, USA), quantified and stored at −80 °C. Whole-cell currents from oocytes were recorded using an OC-725C amplifier (Warner Instruments, Hampden, CT, USA) coupled to Patchmaster software (Heka Elektronik, Lambrecht, Germany). Data analyses were done with Excel, Igor Pro and GraphPad Prism v.5 and the results were shown as the mean ± SEM. Statistical comparisons were done using two-way ANOVA followed by the Bonferroni test, with p < 0.05 indicating significance.

## Figures and Tables

**Figure 1 toxins-11-00335-f001:**
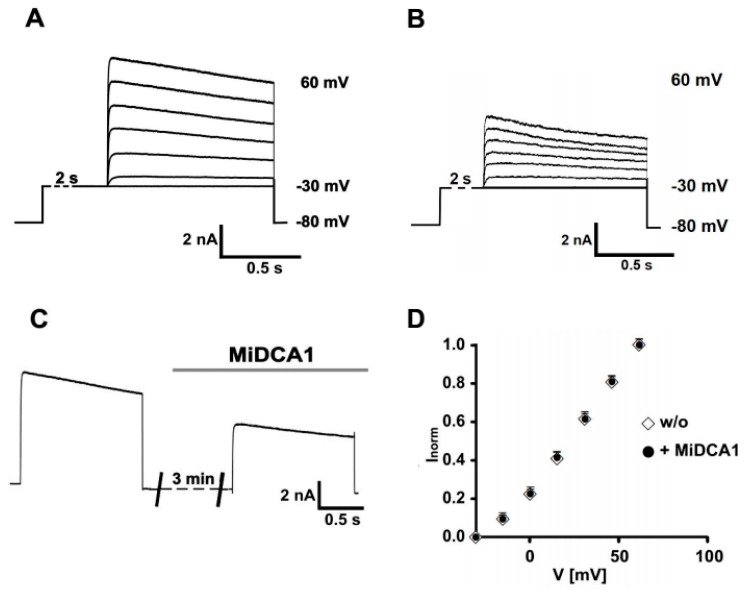
The sensitivity of whole-cell dorsal root ganglion (DRG) potassium currents to MiDCA1. (**A**) Representative control and (**B**) MiDCA1 (1 µM)-treated current traces of DRG potassium currents evoked by 1 s voltage steps to test potentials between −30 mV and +60 mV in 15 mV increments using an EPC9 patch-clamp amplifier combined with PULSE software (HEKA Elektronik, Lambrecht, Germany). DRG neurons were superfused at a flow rate of 1 mL/min with an external solution containing (in mM): NaCl 150, KCl 5, CaCl_2_ 2.5, MgCl_2_ 2, HEPES 10, and D-glucose 10, adjusted to pH 7.4 with NaOH. The pipette solution was (in mM): KCl 140, CaCl_2_ 1, MgCl_2_ 2, EGTA 9, HEPES 10, Mg-ATP 4, and GTP (Tris salt) 0.3, adjusted to pH 7.4 with KOH. Signals were filtered at 0.2–4 kHz with low pass Bessel characteristics, amplified as required and digitized at sampling intervals between one ms and 40 ms. The program PULSEFIT (HEKA Elektronik) was used to analyze the current traces. (**C**) Representative DRG outward currents showing their sensitivity to MiDCA1. The toxin was applied three min prior to the recordings. The gray horizontal bar in (**C**) indicates the duration of MiDCA1 application during the current recording. (**D**) Normalized outward current amplitudes (Inorm) measured before (white diamonds—w/o) and after (black circles) the application of 1 µM MiDCA1 were plotted against test voltages. Current amplitudes recorded from MiDCA1-sensitive neurons before and three min after MiDCA1 application as shown in (**C**) were subtracted from each other to obtain information about the MiDCA1-sensitive current. The points represent the mean ± SEM (*n* = 19). All measurements were done at 37 °C. Vertical and horizontal scale bars in (**A**–**C**) indicate current amplitude and pulse duration, respectively.

**Figure 2 toxins-11-00335-f002:**
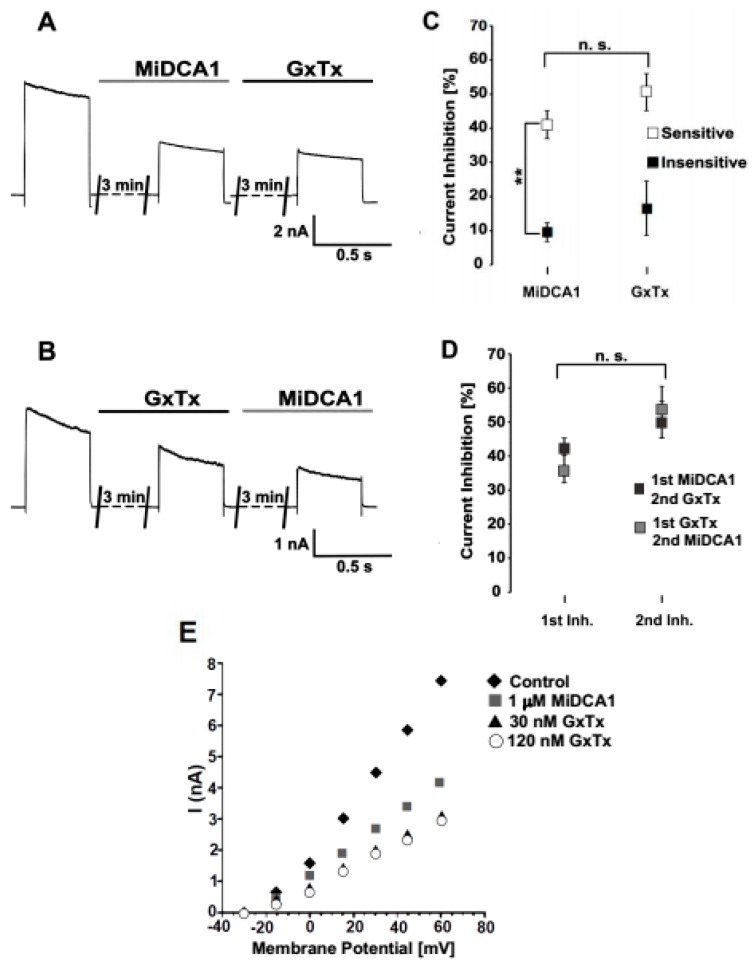
Comparison of the inhibitory activity of MiDCA1 and guangxitoxin (GxTx) on DRG potassium currents. (**A**) Representative recordings of DRG potassium currents after application of 1 µM MiDCA1 (gray horizontal bar) and subsequent application of 30 nM GxTx (black horizontal bar). (**B**) Representative recordings of DRG potassium currents after application of 1 µM MiDCA1 (gray horizontal bar) preceded by application of 30 nM GxTx (black horizontal bar). Currents were elicited by a 1 s voltage step to +60 mV from a holding potential of −30 mV. Toxins were applied three min prior to the recordings. All measurements were done at 37 °C. (**C**) Data were acquired using the experimental design shown in (**A**,**B**). Relative current inhibition was obtained by dividing plateau current amplitudes, recorded after toxin application at the end of a 1 s test pulse to +60 mV, by the current amplitude recorded before application. White rectangles represent the data for toxin-sensitive DRG neurons; black rectangles represent those of DRG-insensitive neurons. (**D**) Relative current inhibition, calculated as in (**C**), measured after MiDCA1 application followed by GxTx application (1st MiDCA1, 2nd GxTx—black rectangles) or after GxTx application followed by MiDCA1 application (1st GxTx, 2nd MiDCA1—gray rectangles). (**E**) Outward current amplitudes measured before (black diamonds—Control) and after the application of 1 μM MiDCA (gray rectangles), 30 nM GxTx (black triangles), and 120 nM GxTx (white circles) plotted against the test voltages. Note that the response to 120 nM GxTx was virtually identical to that seen with 30 nM GxTx. The results in panels (**C**,**D**) are shown as the mean ± SEM (*n* = 7). n.s. not significant (panels C and D).

**Figure 3 toxins-11-00335-f003:**
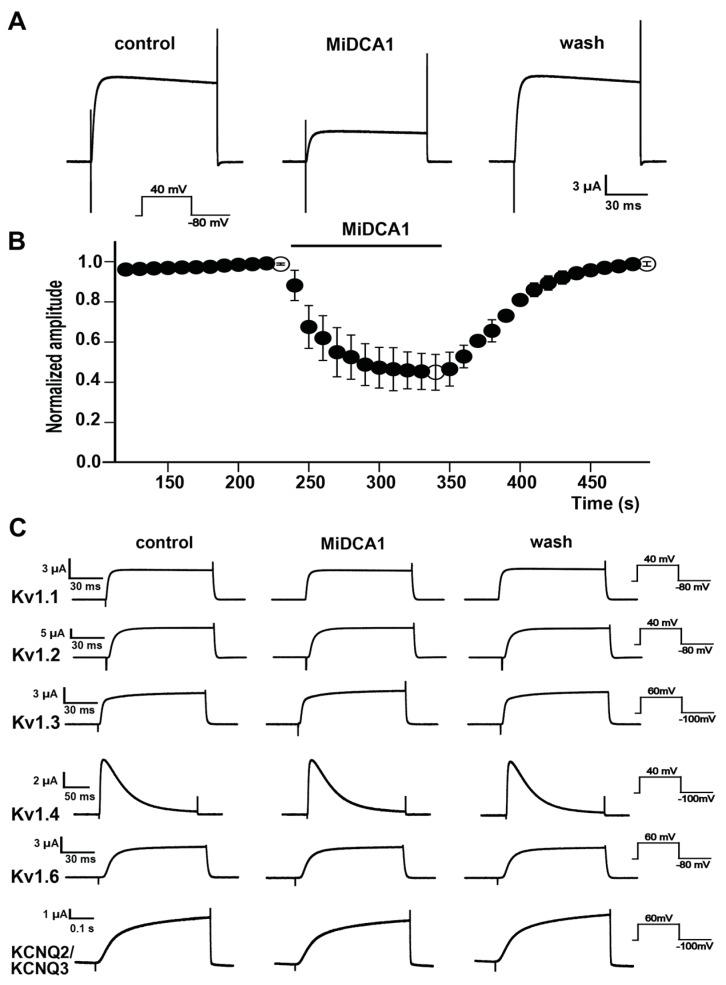
Inhibition of Kv2.1 channels by MiDCA1. (**A**) Representative current traces of the Kv2.1 channel recorded from *Xenopus* oocytes before (control), during incubation with 1 μM MiDCA1, and during washout (wash). The current traces correspond to the points shown in (**B**) as open circles. (**B**) Normalized current amplitude before (control), during application of 1 µM MiDCA1 (horizontal bar), and during washout (wash). Inhibition of current was calculated as the difference between the first and second open circles (0.55 ± 0.089, *n* = 3, *p* < 0.001). (**C**) Typical current traces from several types of Kv channels expressed in oocytes and screened for inhibition by MiDCA1. Symbols are identical to those in (**A**). Oocytes were transfected with the Kv channels as described in the Methods, and at least three cells were tested for each channel type, with no significant differences in the responses within each set of cells. None of the channel types screened was sensitive to blockade by MiDCA1, except for Kv2.1 (panels **A** and **B**).

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
