# Peer review of "Inhibition of Kv2.1 Potassium Channels by MiDCA1, A Pre-Synaptically Active PLA2-Type Toxin from Micrurus dumerilii carinicauda Coral Snake Venom"

_toxins, 2019, doi:10.3390/toxins11060335_

Round 1

Reviewer 1 Report

The authors of “Selective inhibition of Kv2 potassium channels by MiDCA1, a presynsptic PLA2-type toxin from M.d.c” has described a dual function of MiDCA1 including phospholipase A2 activity and potassium channel subtype Kv2 activity. The authors assessed the activity on Kv2 channels in DRG and compared it to a known Kv2 inhibitor, spider toxin Guangxitoxin. They identified that MiDCA1 inhibits 41% of potassium channels in DRGs at 30 nM. The study is interesting and well carried out, but even for a communication, it is a bit brief and would benefit from some additional experiments and information to improve the article.  

Please see comments on the paper below:

1)    There is no mention anywhere on how and from where MiDCA1 was acquired, nor how it was quantified prior to use.

2)    Why did the authors choose 30 nM as the concentration tested? If only 41% inhibition is observed at 30 nM, wouldn’t you test a higher dose?

3)    Confirmation that Kv2 is indeed the target by using heterologous expression in oocytes or stably mammalian transfected cells would be desired to confirm activity at the particular subtype since it is not known what subunit composition Kv2 has in DRGs.

4)    The title includes the word selective, suggesting that various Kv subtypes have been tested, but this is not included.

5)    Further subtype selectivity testing would have also improved understanding as to why there was improvement in inhibition depending on the order of toxin addition.

Minor points

1)    The paper needs another edit/proofread as there are spaces missing between 9 and week on page 1 line 34, spaces missing between 37 and ºC etc.

2)    In Figure 2, the symbols need to be revised as when printed in black and white, both red and black looks black and can’t be distinguished.

Author Response

Replies to Reviewer 1

Dear Reviewer,

Thank you for taking the time to examine our work. Detailed replies to your comments are provided below and corresponding alterations to the text are highlighted in green. We hope that we have satisfactorily addressed your concerns.

Sincerely,

Cháriston A. Dal Belo, PhD

Comments:

The authors of “Selective inhibition of Kv2 potassium channels by MiDCA1, a presynsptic PLA2-type toxin from M.d.c” has described a dual function of MiDCA1 including phospholipase A2 activity and potassium channel subtype Kv2 activity. The authors assessed the activity on Kv2 channels in DRG and compared it to a known Kv2 inhibitor, spider toxin Guangxitoxin. They identified that MiDCA1 inhibits 41% of potassium channels in DRGs at 30 nM.

General comment: The study is interesting and well carried out, but even for a communication, it is a bit brief and would benefit from some additional experiments and information to improve the article.  Please see comments on the paper below.

Reply: As requested, we have provided additional information and data to support our principal conclusion that MiDCA1 acts by selectively blocking Kv2.1 channels. Several of these inclusions are indicated in our replies to the reviewer´s comments below.

Comment 1: There is no mention anywhere on how and from where MiDCA1 was acquired, nor how it was quantified prior to use.

Reply: As requested, we have now mentioned the source of MiDCA1 in the Methods section (lines 172-173). The toxin was purified from M. d. carinicauda venom using RP-HPLC as described by Dal Belo et al. (2005) – reference 13 in the manuscript reference list. This information was included in the legend for Fig. 1 in the original version but has now been transferred to the Methods section.

Reference:

Dal Belo CA et al. (2005) Pharmacological and structural characterization of a novel phospholipase A2 from Micrurus dumerilii carinicauda venom. Toxicon 46, 736-750.

Comment 2: Why did the authors choose 30 nM as the concentration tested? If only 41% inhibition is observed at 30 nM, wouldn’t you test a higher dose?

Reply: Guangxitoxin (GxTx) inhibits Kv2.1 channels in the nanomolar range, e.g., ~40 nM (Herrington et al., 2006) and, in our experimental conditions, evoked currents of Kv2.1 channels expressed in Xenopus oocytes were completely inhibited by ≤30 nM GxTx (data not shown). We have now included results showing that a higher concentration of GxTx (120 nM) did not cause additional blockade compared to 30 nM (see text, lines 105-108 and Fig. 2E and corresponding legend, lines 128-131, underlined in blue). These findings suggest that the remaining current is probably mediated by other Kv channels that are not blocked by GxTx.

Reference:

Herrington et al. (2006) Blockers of the delayed-rectifier potassium current in pancreatic b-cells enhance glucose-dependent insulin secretion. Diabetes 55, 1034-1042.

Comment 3: Confirmation that Kv2 is indeed the target by using heterologous expression in oocytes or stably mammalian transfected cells would be desired to confirm activity at the particular subtype since it is not known what subunit composition Kv2 has in DRGs.

Reply: As requested, we have now included results with other Kv channel subtypes expressed in Xenopus oocytes and incubated with MiDCA1. These findings are now shown in Fig. 3 and corresponding legend as well as in the main text (lines 143-151).

Comment 4: The title includes the word selective, suggesting that various Kv subtypes have been tested, but this is not included.

Reply: In view of our results with a variety of Kv channel (sub)types (Kv1.1, Kv1.2, Kv1.3, Kv1.4, Kv1.6 and KCNQ2/KCNQ3) screened for inhibition by MiDCA1, in addition to Kv2.1 (see reply to Comment 3 above), we have retained the word ‘selective’ in the title as we feel that our findings support the use of this term.

Comment 5: Further subtype selectivity testing would have also improved understanding as to why there was improvement in inhibition depending on the order of toxin addition.

Reply: We agree that such additional testing would be interesting. However, the difficulty in obtaining more toxin precludes such an analysis at this time (Sigma Chemical Co, from which we originally obtained the venom, no longer sells this venom).

Comment 6: In Figure 2, the symbols need to be revised as when printed in black and white, both red and black looks black and can’t be distinguished.

Reply: We have changed the symbols to black and white (closed and open) (Fig. 2C) or grey and black (Fig. 2D) to make them clearer.

Reviewer 2 Report

Figure 1 - I think the authors should show those currents that are blocked by MiDCA1, rather than the IV of the current that remains (or as well as)

The authors should aim to provide a concentration response curve for MiDCA1.

30 nM GxTx may not be quite enough to block all the Kv2 (Fig 2B). Probably need a bit more toxin in this experiment. I would suppose this would explain why MiDCA1 causes a little further block of the K+ currents

Author Response

Replies to Reviewer 2

Dear Reviewer,

Thank you for taking the time to examine our work. Detailed replies to your comments are provided below and corresponding alterations to the text are blue. We hope that we have satisfactorily addressed your concerns.

Sincerely,

Cháriston A. Dal Belo, PhD

Comments:

Comment 1: Figure 1 - I think the authors should show those currents that are blocked by MiDCA1, rather than the IV of the current that remains (or as well as).

Reply: As requested, we have now included these currents in Fig. 1B.  

Comment 2: The authors should aim to provide a concentration response curve for MiDCA1.

Reply: Because of the limited amount of toxin available, we did not do a complete concentration-response curve for MiDCA1 in DRG neurons. However, in our initial pharmacological characterization of MiDCA1, we used a higher concentration (2.4 mM) than that used here and there was apparently no marked difference in the ability of these two concentrations to block potassium currents.

Reference:

Dal Belo CA et al. (2005) Pharmacological and structural characterization of a novel phospholipase A2 from Micrurus dumerilii carinicauda venom. Toxicon 46, 736-750.

Comment 3: 30 nM GxTx may not be quite enough to block all the Kv2 (Fig 2B). Probably need a bit more toxin in this experiment. I would suppose this would explain why MiDCA1 causes a little further block of the K+ currents

Reply: Guangxitoxin (GxTx) inhibits Kv2.1 channels in the nanomolar range, e.g., ~40 nM (Herrington et al., 2006) and, in our experimental conditions, evoked currents of Kv2.1 channels expressed in Xenopus oocytes were completely inhibited by ≤30 nM GxTx (data not shown). We have now included results (an I/V curve) showing that a higher concentration of GxTx (120 nM) did not cause additional blockade compared to 30 nM (see text, lines 105-108 and Fig. 2E and corresponding legend, lines 128-131, underlined in blue). These findings suggest that the remaining current is probably mediated by other Kv channels that are not blocked by GxTx.

Reference:

Herrington et al. (2006) Blockers of the delayed-rectifier potassium current in pancreatic b-cells enhance glucose-dependent insulin secretion. Diabetes 55, 1034-1042.

Round 2

Reviewer 1 Report

The authors have addressed the comments, and included additional experiments strengthening the manuscript. I would still urge caution in using the word selective as even though several additional Kv channels have been tested, there are still plenty Kv channels left untested. Language in the form of " the peptide showed selectivity for Kv2.1 over all the other Kv channels tested in this study" would be more appropriate. 

Author Response

Dear Reviewer,

Thank you for taking the time to examine our work once again. A reply to your comment is provided below and corresponding alterations to the text are highlighted in green. We hope that we have satisfactorily addressed your concerns.

Sincerely,

Cháriston A. Dal Belo, PhD

Comments:

The authors have addressed the comments, and included additional experiments strengthening the manuscript. I would still urge caution in using the word selective as even though several additional Kv channels have been tested, there are still plenty Kv channels left untested. Language in the form of "the peptide showed selectivity for Kv2.1 over all the other Kv channels tested in this study" would be more appropriate.

Reply: The text of the manuscript has been altered as suggested, with the alterations to the manuscript text indicated in green in the Abstract (page 1, lines 12-14) and Page 6, lines 154-157. We also decided to delete the word “Selective” from the title of the manuscript.
